# Hybrid Dynamic High-Order Functional Correlations and Divisive Normalization for Improved Classification of Schizophrenia and Bipolar Disorder

**Qiang Li**    **Vince D. Calhoun**

Tri-Institutional Center for Translational Research in Neuroimaging and Data Science (TReNDS)
Georgia State, Georgia Tech, Emory, Atlanta, GA, USA
`qli27@gsu.edu`

## Abstract

Schizophrenia and bipolar disorder are devastating psychiatric disorders that can be difficult to adequately classify, considering commonalities that make it difficult to distinguish between them using conventional classification approaches based on low-order functional connectivity. Recently, high-order functional connectivity, which usually refers to interactions beyond pairwise connections, has emerged as a promising method for diagnosing psychiatric illnesses. This research applied multiple strategies to distinguish between schizophrenia and bipolar disorder using features derived from dynamic high-order functional connectivity and divisive normalization. The approach that produced the greatest results combined dynamic high-order functional correlations and divisive normalization to examine patterns of intrinsic connection time courses collected from resting-state fMRI. Our findings indicate that resting-state fMRI-based dynamic high-order functional connectivity and feature enhancement through divisive normalization classification hold significant promise for improving the accuracy of psychiatric diagnoses. Moreover, to the best of our knowledge, this study is the first to integrate divisive normalization with functional connectivity in fMRI.

## 1   Introduction

The use of resting-state functional magnetic resonance imaging (rsfMRI) to identify individuals with schizophrenia (SZ) and bipolar disorder (BP) has gained substantial attention in global medicine and clinical practice [1–5]. rsfMRI allows us to investigate the intricate interaction of the brain's functional connections, revealing identifiable patterns of neural activity that may shed light on the specific characteristics of various mental disorders.

Functional connectivity elucidates the intricate interplay and communication among different brain regions, coordinating various cognitive processes. Although both low-order and high-order (interactions beyond pairwise connections) static and dynamic functional connectivity are essential for our daily cognitive functioning, recent research suggests that high-order functional connectivity (HOFC) may have a distinct advantage in terms of increasing cognitive flexibility and optimizing task performance [6–9]. Meanwhile, emerging evidence highlights the superior efficacy of high-order functional connectivity in fostering cognitive adaptability and diagnosing brain dysfunction from an information theory perspective [10–17]. To enhance the feature from high-order dynamic functional correlations, we incorporate a divisive normalization strategy to further enhance sensitivity based on work showing its potential for capturing neural computation in the visual cortex and its use in the computer vision field [18–21].

Proceedings of the II edition of the Workshop on Unifying Representations in Neural Models (NeurIPS 2024).

As we show, the classification of SZ and BP patients considering high-order functional connectivity appears to be a promising route, with the potential to improve the precision and specificity of psychiatric diagnostic procedures. This advancement, in turn, holds the potential to usher in an era of personalized and precisely targeted interventions, ultimately optimizing the care and outcomes for individuals grappling with these complex mental health challenges.

Within the scope of this study, our primary objective is to discriminate between individuals with SZ and those with BP using high-order functional connectivity. To achieve this, we employ a combination of static and dynamic approaches involving high-order complex brain networks. Our findings can be summarized into two key observations. Firstly, we demonstrate that dynamic high-order functional connectivity furnishes a more information-rich foundation compared to low-order functional connectivity. Secondly, we illustrate that the integration of dynamic high-order brain networks and divisive normalization significantly enhances classification accuracy. Therefore, we suggest considering the potential of dynamic high-order functional connectivity as a metric to aid in the precise diagnosis of psychiatric disorders.

## 2 Materials and Methods

### 2.1 rsfMRI Dataset Acquisition

The entirety of our dataset, comprising derivative information from 471 subjects meticulously selected from the Bipolar and Schizophrenia Network for Intermediate Phenotypes (B-SNIP) consortium [22], serves as the fundamental cornerstone of this research endeavor. The 288 subjects represented typical SC, and 183 were diagnosed with BP. The open-source data can be downloaded from `https://www.kaggle.com/competitions/psychosis-classification-with-rsfmri/data`.

### 2.2 rsfMRI Dataset Processing

The rigorous preprocessing pipeline applied to rsfMRI data, illustrated in Fig.1, encompasses the following essential steps:

The rsfMRI data of each participant underwent a standardized preprocessing protocol, inclusive of critical steps such as rigid body motion correction, slice timing correction, and distortion correction. Following this, the preprocessed subject data were registered into a shared spatial template, resampled to isotropic voxels of $3mm^3$, and then underwent spatial smoothing utilizing a Gaussian kernel with a full width at half-maximum of $6mm$. Utilizing a multi-spatial-scale template comprising 105 intrinsic connectivity networks (ICNs), which in itself was derived from an extensive dataset of over 100K subjects called the *Neuromark_fMRI_2.1 network template* [23], which can be downloaded from `https://trendscenter.org/data/`. Meanwhile, subject-specific ICN time courses were extracted employing a spatially constrained independent component analysis (scICA) approach, implemented using the GIFT software toolbox (http://trendscenter.org/software/gift).

These extracted time courses underwent a thorough cleaning process in accordance with established standards [23]. This meticulous and precision-driven methodology serves as the bedrock for upholding the integrity and trustworthiness of our dataset, thereby laying the foundation for subsequent precision analyses. Our dataset leverages the power of 105 intrinsic connectivity networks (ICNs), as visually depicted in Fig.2. These 105 ICNs are categorized into six distinct domains based on structural and functional characteristics: visual domain (VI, 12 sub-networks), cerebellar domain (CB, 13 sub-networks), temporal-parietal domain (TP, 13 sub-networks), sub-cortical domain (SC, 23 sub-networks), sensorimotor domain (SM, 13 sub-networks), and high-level cognitive domain (HC, 31 sub-networks). Moreover, these ICNs were derived through a multi-spatial-scale, scICA methodology, a technique that enhances generalizability and facilitates the comparability of results across diverse studies [24].

### 2.3 Low-Order Functional Connectivity (LOFC)

Before we go into HOFC, here we will quickly introduce the most common low-order functional connectivity (LOFC), which is estimated based on Pearson correlation (PC).

**A. Data-Driven Brain Network Reference Estimation-Group Spatial ICA**

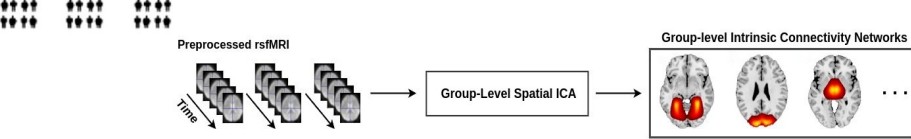

**B. Reference-Driven Brain Networks Estimation - Spatially Constrained ICA**

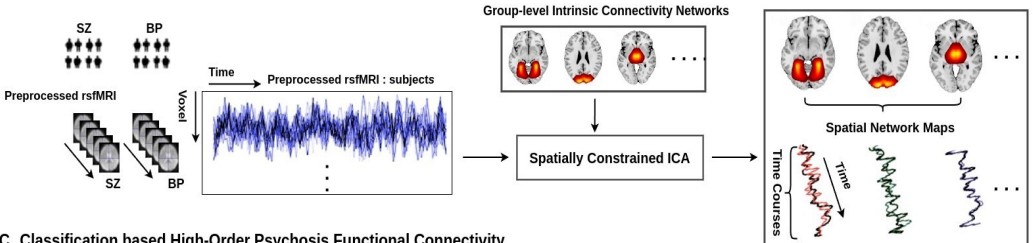

**C. Classification based High-Order Psychosis Functional Connectivity**

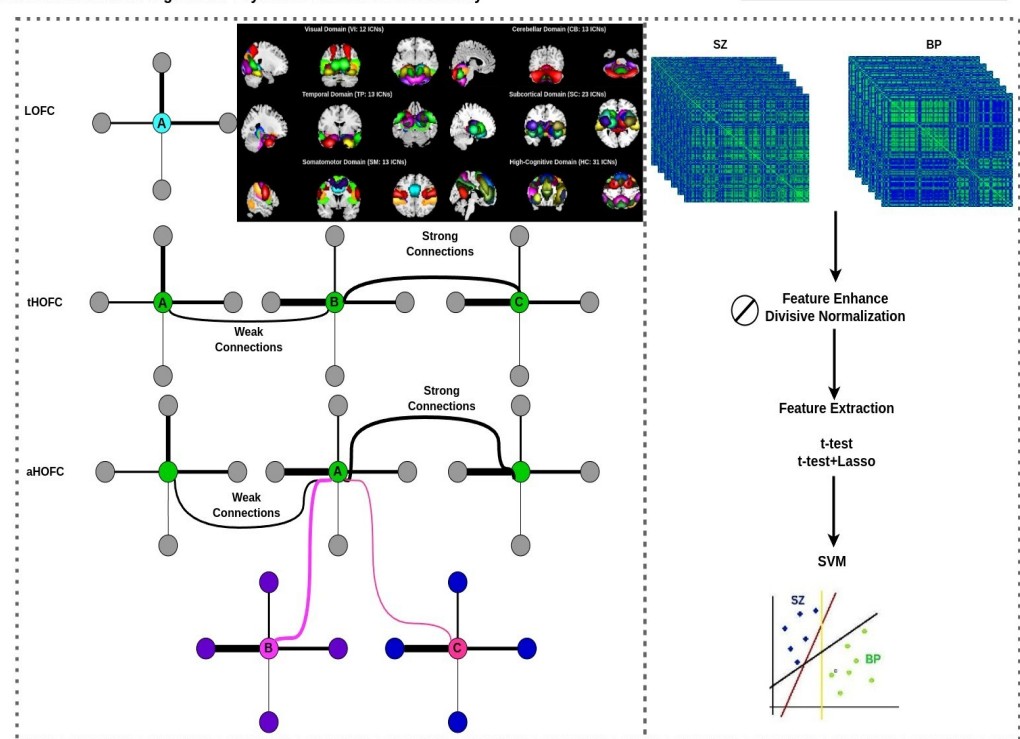

Figure 1: The framework designed for classifying SZ and BP. **A.** The data-driven brain network references were estimated from a substantial cohort of subjects through group-level spatial ICA. **B.** The spatial network and corresponding time courses were constructed utilizing subjects with SZ and BP employing spatially constrained ICA, based on previously established group-level intrinsic connectivity networks. **C.** The construction of low-order and high-order functional connectivity (LOFC vs. HOFC) matrices (i.e., topographical profile similarity-based HOFC, tHOFC; associated HOFC, aHOFC) was presented in the left panel. Divisive normalization was subsequently applied to these functional connectivity matrices to enhance their features. They were then fed into the machine learning model to extract relevant features using t-test and t-test plus Lasso, with the goal of maximizing classification results.

Letting $x_i(t)$ and $x_j(t)$ represent the rsfMRI signals for two brain regions $i$ and $j$ at time point $t(t = 1, \ldots, T)$, then LOFC based on PC be denoted as,

$$\rho_{ij} = \frac{\sum_{t=1}^{T} \left(x_i(t) - \hat{x_i}\right)\left(x_j(t) - \hat{x_j}\right)}{\sqrt{\sum_{t=1}^{T} \left(x_i(t) - \hat{x_i}\right)^2 \sum_{t=1}^{T} \left(x_j(t) - \hat{x_j}\right)^2}} \quad (1)$$

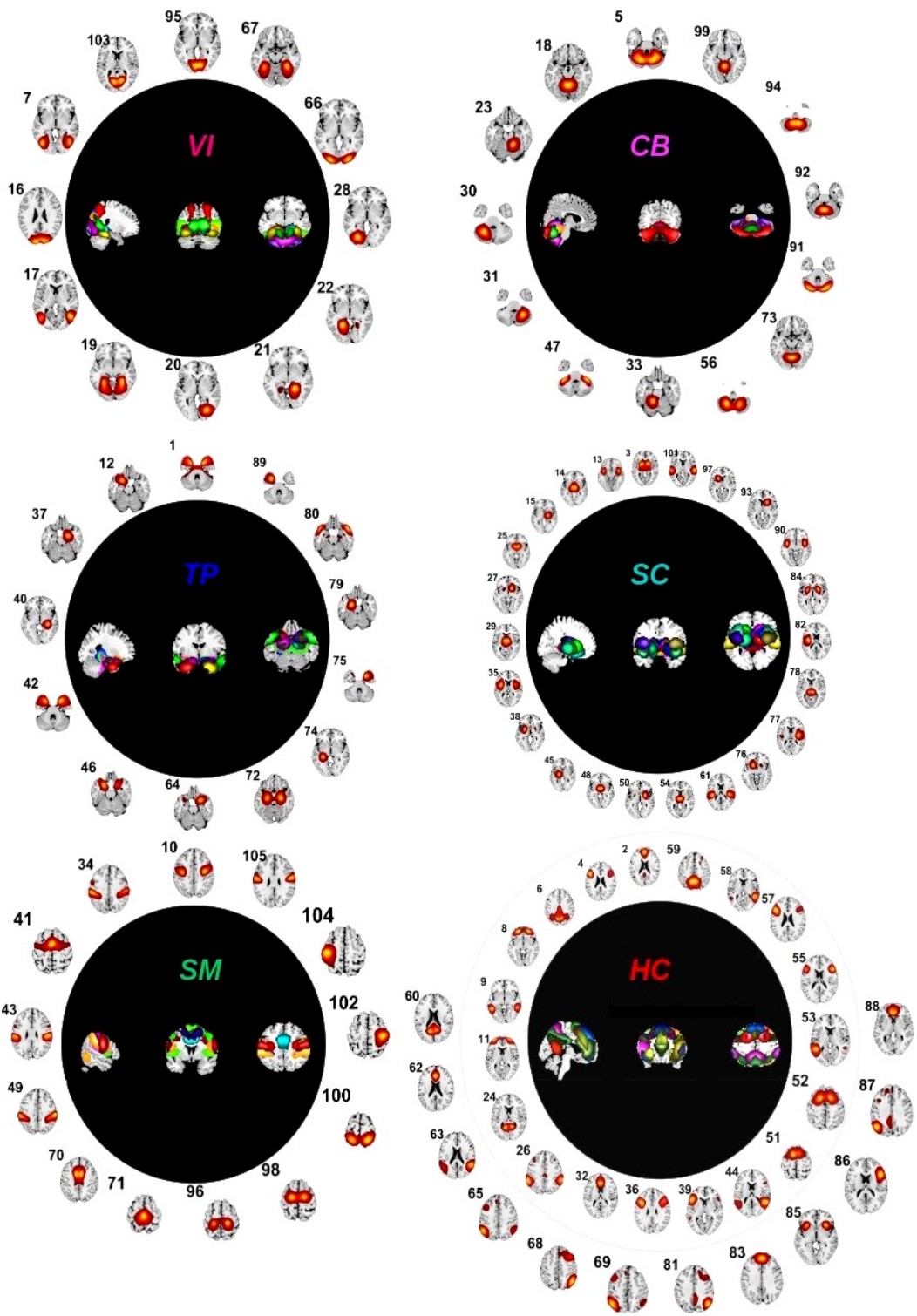

Figure 2: A total of 105 intrinsic brain networks were categorized into six domains using the *Neuromark_fMRI_2.1 network template* [23], available at https://trendscenter.org/data/. These domains include: visual (12 ICNs), cerebellar (13 ICNs), temporal (13 ICNs), subcortical (23 ICNs), somatomotor (13 ICNs), and high cognitive (31 ICNs).

where $\hat{x}_i$ and $\hat{x}_j$ are the average of the rsfMRI signals at regions $i$ and $j$.

## 2.4 High-Order Functional Connectivity (HOFC)

According to the proposal by Han et al. [25, 26], the pairwise connection represented by tHOFC and aHOFC differs from that of LOFC due to variations in input signals. Nonetheless, both tHOFC and aHOFC characterize the interaction between two brain regions, establishing high-order functional connectivity. The LOFC are derived from the rsfMRI time series. However, these time series serve as local LOFC profiles for tHOFC. Notably, a noticeable variation between LOFC and tHOFC/aHOFC may exist for the same two brain regions, attributed to differences in input signals [25–27].

### 2.4.1 Topological High-Order Functional Connectivity (tHOFC)

Letting $\rho_i$ represents the LOFC profile for region $i$, and $\rho_{i\cdot} = \{\rho_{ik|k\in\mathbb{R},k\neq i}\}$. Then the tHOFC can be denoted as,

$$t_{ij} = \frac{\sum_k (\rho_{ik} - \hat{\rho}_i\cdot)(\rho_{jk} - \hat{\rho}_j\cdot)}{\sqrt{\sum_k (\rho_{ik} - \hat{\rho}_i\cdot)^2}\sqrt{\sum_k (\rho_{jk} - \hat{\rho}_j\cdot)^2}} \qquad (2)$$

### 2.4.2 Associated High-Order Functional Connectivity (aHOFC)

Similar to the calculation of tHOFC, aHOFC will be calculated as follows:

$$\alpha_{ij} = \frac{\sum_k (t_{ik} - \hat{t_{i\cdot}})(\rho_{jk} - \hat{\rho_{j\cdot}}^2)}{\sqrt{\sum_k (t_{ik} - \hat{t}_i\cdot)^2}\sqrt{\sum_k (\rho_{jk} - \hat{\rho}_j\cdot)^2}} \qquad (3)$$

In Fig.3, we present LOFC, tHOFC, and aHOFC. It is evident that HOFC significantly improves connectivity information compared to LOFC, while also preserving the original topological structure information.

## 2.5 Dynamic High-Order Functional Connectivity (dHOFC)

The dHOFC is calculated based on dynamic, time-varying LOFC profiles. First, we estimate dLOFC based on Pearson correlation for time-varying time series as follows:

$$d\rho_{il}(\tau) = \frac{\sum_{t=\tau}^{\tau+\omega-1}(x_i(t) - \hat{x_i^\tau})(x_l(t) - \hat{x_l^\tau})}{\sqrt{\sum_{t=\tau}^{\tau+\omega-1}(x_i(t) - \hat{x_i^\tau})^2}\sqrt{\sum_{t=\tau}^{\tau+\omega-1}(x_l(t) - \hat{x_l^\tau})^2}} \qquad (4)$$

where $\tau = 1,\ldots,T-\omega+1; i,l\in\mathbb{R}, i\neq l$, $\tau$ indicates time, $\omega$ refers to window length, and $\hat{x_i^\tau}$ refers to the average value of the segment of the rsfMRI signal starting from $\tau$. and then, dHOFC is constructed based on dLOFC profiles,

$$d\boldsymbol{\rho}_{il,jk} =$$
$$\frac{\sum_{\tau=1}^{T-\omega+1}\left(d\rho_{\mathrm{il}}(\tau) - \hat{d\rho_{\mathrm{il}}}\right)\left(d\rho_{\mathrm{jk}}(\tau) - \hat{d\rho_{\mathrm{jk}}}\right)}{\sqrt{\sum_{\tau=1}^{T-\omega+1}\left(d\rho_{\mathrm{il}}(\tau) - \hat{d\rho_{\mathrm{il}}}\right)^2}\sqrt{\sum_{\tau=1}^{T-\omega+1}\left(d\rho_{\mathrm{jk}}(\tau) - \hat{d\rho_{\mathrm{jk}}}\right)^2}} \qquad (5)$$

According to the criteria given above, dHOFC provides more connectomic information compared to LOFC, tHOFC, and aHOFC. The HOFC algorithms were implemented using the BrainNetClass toolbox (https://github.com/zzstefan/BrainNetClass).

## 2.6 Divisive Normalization

Divisive normalization is a phenomenological model that captures the nonlinear response properties observed widely across sensory cortical areas [18]. Here, divisive normalization was employed to enhance features in functional connectivity matrices, utilizing a standard neural computational way in the human primary visual cortex, which has since been widely applied in image processing and deep learning [18–21]. Here, we adapted and applied it to functional connectivity matrices, and it can be mathematically represented as follows:

$$\left(\text{sign}(d\boldsymbol{\rho}_{il,jk}) \cdot \kappa \cdot \frac{|d\boldsymbol{\rho}_{il,jk}|^{\gamma}}{b + |d\boldsymbol{\rho}_{il,jk}|^{\gamma}}\right)^{-\frac{1}{2}} \tag{6}$$

There are several hyperparameters (i.e., $\kappa$, $\gamma$, $b$), and after tuning parameters, we selected $\gamma = 2$, $b = 0.0625$, and $\kappa = 0.5$. Here $d\boldsymbol{\rho}_{il,jk}$ refers to dHOFC matrix.

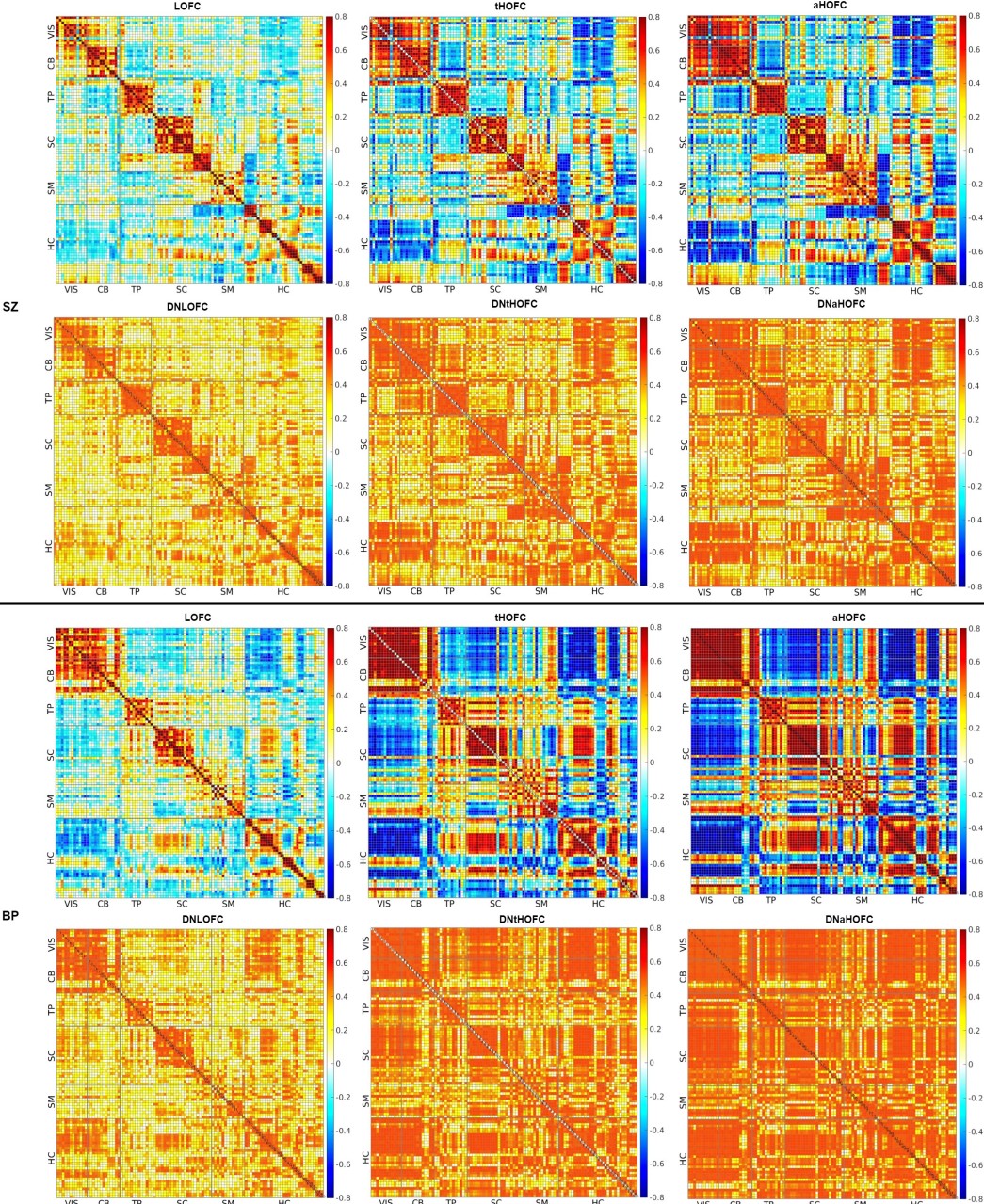

Figure 3: Functional connectivity matrices were constructed from SZ and BP subjects. The first row shows LOFC, tHOFC, and aHOFC, while the second row displays each after applying divisive normalization for the SZ subjects. The third row shows LOFC, tHOFC, and aHOFC, while the fourth row displays each after applying divisive normalization for the BP subjects. The community patterns present in the functional connectivity matrices were labeled accordingly.

## 2.7 Feature Extraction

The t-test and Least absolute shrinkage and selection operator (Lasso) were used for feature extraction. Assuming $X$ is the functional connectivity matrix of features (with shape $n \times p$, where $n$ is number of samples, and $p$ is the number of features), $y$ is the vector of class labels. For each feature $j$, the t-test will be,

$$t_j = \frac{\bar{X}_{j1} - \bar{X}_{j2}}{s_j / \sqrt{n_j}} \tag{7}$$

where $\bar{X}_{j1}$ and $\bar{X}_{j2}$ are the means of feature $j$ in the two classes. $s_j$ is the standard deviation of feature $j$, and $n_j$ is the number of samples in class $j$.

Compute the corresponding p-value $p_j$ for each $t_j$, which indicates the probability of obtaining the observed t-statistic under the null hypothesis (that the feature is not associated with the class labels). Features with p-values sufficiently low (typically below a threshold, e.g., 0.05) are then selected as statistically significant features. The Lasso is a regularization technique that performs both feature selection and regularization to improve the prediction accuracy [28].

## 2.8 Evaluation metrics

Let PL represent the total number of SZ functional connectivity matrices, NL represent the total number of BP functional connectivity matrices, TP represent the true positives (number of SZ functional connectivity matrices correctly identified as SZ), FP represent the false positives (number of BP functional connectivity matrices incorrectly identified as SZ), TN represent the true negatives (number of BP functional connectivity matrices correctly identified as BP), and FN represent the false negatives (number of SZ functional connectivity matrices incorrectly identified as BP).

- F1-score

$$\frac{2TP}{2TP + FP + FN} \tag{8}$$

The receiver operating characteristic (ROC) curve is a probability curve created by plotting the true positive (TP) rate against the false positive (FP) rate at various threshold settings. The area under the ROC curve (AUC) represents the performance measure of a classifier. A higher AUC indicates that the classifier is better at distinguishing between SZ and BP.

# 3 Results

To examine the differences between SZ and BP is complicated because the two disorders share many features. Classifying SZ and BP using machine learning relies heavily on the extracted features. In this study, we extracted features from both low-order and high-order functional connectivity metrics. Divisive normalization was employed to enhance these features, followed by feature selection using t-test and Lasso to maximize classification accuracy. First and foremost, we observed that high-order functional connectivity presents distinct connectivity patterns compared to low-order functional connectivity in SZ and BP, as illustrated in Fig.3. The strongest connection pattern between HC and SC exists in BP compared to SZ, and these differences could have implications for understanding the underlying mechanisms of BP and its impact on high-level cognitive functions. Moreover, they may serve as a biomarker to differentiate between SZ and BP. Secondly, we found that dynamic high-order functional connectivity indeed gives us better classification results (*86.27%*) compared to baseline (*64.41%*) and even deep learning results (*70.50%*), as shown in Tab.1. As we mentioned before, one of the major reasons is that dynamic high-order functional connectivity capture additional information relative to other approaches. This, typically ignored, higher order information may be important for us to better understand complex brain functions. Thirdly, we observed that divisive normalization significantly improved classification accuracy compared to not employing divisive normalization (*96.61%* vs. *86.27%*). These results demonstrate that divisive normalization is essential for feature enhancement and significantly improves classification performance. To our knowledge, this study is the first to integrate divisive normalization with functional connectivity for this purpose.

Table 1: Performance Comparison of SZ and BP Classifications: Utilizing a Kaggle baseline, we selected the best classification result, and also included other approaches for comparison with our proposed method.

| Methods\Metrics | F1-score | AUC |
|---|---|---|
| **Baseline** | | |
| Kaggle Best Score: `https://www.kaggle.com/competitions/psychosis-classification-with-rsfmri/leaderboard` | - | **64.41%** |
| **Deep Learning - CNN (1D & 3D) - Best Score** | | |
| **1D CNN (parameter required)** | | |
| Time Series/raw ICN [29] | - | **70.50%** |
| **3D CNN (parameter required)** | | |
| Time Series/Scalograms and Spectrograms [29] | - | **62.60%** |
| **Machine Learning - SVM (2D Spatial Domain)** | | |
| **Static Low-Order Functional Connectivity (no parameter required)** | | |
| PC/Connection coefficients/LASSO ($\lambda = 0.05$)/Leave-one-out cross validation | 51.80% | **39.56%** |
| PC/Connection coefficients/T-test + LASSO/Leave-one-out cross validation | 58.14% | **48.27%** |
| PC/Local clustering coefficients/LASSO ($\lambda = 0.05$)/Leave-one-out cross validation | 70.64% | **62.64%** |
| PC/Local clustering coefficients/T-test + LASSO/Leave-one-out cross validation | 74.40% | **46.61%** |
| **Static High-Order Functional Connectivity (no parameter required)** | | |
| aHOFC/Connection coefficients/LASSO ($\lambda = 0.05$)/Leave-one-out cross validation | 54.51% | **45.92%** |
| aHOFC/Connection coefficients/T-test + LASSO/Leave-one-out cross validation | 58.41% | **47.20%** |
| aHOFC/Local clustering coefficients/LASSO ($\lambda = 0.05$)/Leave-one-out cross validation | 72.05% | **50.24%** |
| aHOFC/Local clustering coefficients/T-test + LASSO/Leave-one-out cross validation | 75.89% | **64.36%** |
| tHOFC/Connection coefficients/LASSO ($\lambda = 0.05$)/Leave-one-out cross validation | 50.87% | **41.74%** |
| tHOFC/Connection coefficients/T-test + LASSO/Leave-one-out cross validation | 57.46% | **50.91%** |
| tHOFC/Local clustering coefficients/LASSO ($\lambda = 0.05$)/Leave-one-out cross validation | 70.52% | **45.06%** |
| tHOFC/Local clustering coefficients/T-test + LASSO/Leave-one-out cross validation | 75.89% | **64.60%** |
| **Dynamic High-Order Functional Connectivity (parameter required)** | | |
| dHOFC/Window length (w=10)/Clustering number (c=50)/ 10-fold cross validation | 26.72% | **86.27%** |
| dHOFC/Window length (w=30)/Clustering number (c=60)/ 10-fold cross validation | 67.45% | **84.64%** |
| **Dynamic High-Order Functional Connectivity (parameter required) + Divisive Normalization** | | |
| dHOFC+DN/Window length (w=10)/Clustering number (c=50)/ 10-fold cross validation | 86.35% | **96.61%** |

# 4    Conclusion

In this study, we observed that high-order functional connectivity generally provides improved classification accuracy compared to low-order functional connectivity. This improvement is primarily due to the additional connection information high-order connectivity offers. Additionally, high-order connectivity better captures the complex, non-pairwise nature of brain information interactions. We investigated various high-order functional connectivity approaches to estimate relationships that extend beyond pairwise correlations. Furthermore, we integrated a divisive normalization model to enhance connection features, which significantly improved classification accuracy. In summary, combining dynamic high-order functional connectivity with divisive normalization proves to be an effective approach for enhancing classification accuracy and holds substantial potential for application in brain disorder diagnosis.

# 5    Future Work

Integrating dynamic high-order functional connectivity and divisive normalization with deep learning models: Future research will focus on combining dynamic high-order functional connectivity and divisive normalization techniques within deep learning frameworks. By leveraging large-scale brain imaging datasets, we aim to develop more sophisticated models that capture intricate patterns of brain connectivity. This approach seeks to enhance the predictive power and generalization of classification algorithms, potentially leading to more accurate diagnoses and a deeper understanding of brain disorders. Implementing these advanced techniques will involve exploring various deep learning architectures, such as convolutional neural networks (CNNs) and recurrent neural networks (RNNs), to effectively process and analyze high-dimensional functional connectivity data [30, 31]. Additionally, we will investigate the integration of temporal dynamics and high-order interactions to capture evolving brain network patterns over time.

Exploring beyond pairwise interactions with advanced topological data analysis: Moving beyond traditional pairwise interactions, future work will investigate advanced topological data analysis techniques to capture more complex relationships within brain connectivity [32–34]. We will explore the use of hypergraphs [34], which can represent higher-order interactions among multiple brain regions, and persistent diagrams [33], which offer insights into the topological features of connectivity patterns across different scales. These methods will enable us to capture and analyze

intricate connectivity structures and dynamics that are not easily represented by pairwise correlations alone. By applying these advanced techniques, we aim to gain a deeper understanding of brain network organization and its implications for neurological and psychiatric disorders.

Identifying potential biomarkers and key network connections in brain disorders: Our future research will focus on identifying potential biomarkers for specific brain disorders by examining which networks or connections between networks are most crucial. This involves analyzing the relationships between different brain regions and their roles in various psychiatric conditions. By leveraging dynamic high-order functional connectivity and divisive normalization, we aim to pinpoint specific networks and their interactions that may serve as reliable biomarkers. Understanding these critical connections will provide insights into the underlying mechanisms of brain disorders and contribute to the development of personalized treatment strategies.

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
