# OpenReview forum: "Hybrid Dynamic High-Order Functional Correlations and Divisive Normalization for Improved Classification of Schizophrenia and Bipolar Disorder"
_NeurIPS.cc/2024/Workshop/UniReps — UniReps_

### Official Review · Reviewer_Axx2 · 2024-09-30
**Interesting idea, but needs a lot of refinement**

**Rating:** 3
**Confidence:** 4

**Review:**

### Summary
This paper tackles the problem of classifying schizophrenia and bipolar disorder based on fMRI data. The authors show that using high-order functional connectivity features allows even simple models such as SVMs to achieve better results compared to the solutions based on the low-order functional connectivity method used so far.

### Major strengths:
* The paper is relevant.
* The idea is interesting and fresh.

### Major weaknesses:
* 2.4 This section seems very unclear to me. What does $ \tau $ represent in Eq. 3? What is even the difference between tHOFC and aHOFC? The paper is designed specifically to solve the domain problem, so it really should be explained better, especially for the AI community.
* Section 2.8 I do not think the description of the metrics is necessary here. I mean especially the formulas.
* The results are very concerning to me. Even though the ROC AUC is higher for the proposed method, sensitivity and specificity measures seem very unbalanced. For instance, 3.78% sensitivity and 95.68% specificity suggest, that the model is virtually incapable of properly recognising schizophrenia at all. Are those results reported for the best thresholds or just for standard 0.5?
* Choosing the Kaggle best score as a baseline does not feel like a good idea. What model did it use? What was the sensitivity or F1 score?

### Minor weaknesses:
* Line 112. I would change "... free parameters" -> "... hyperparameters"
* Eq 1. Should not it be: $p_{ij} = \frac{\sum_{t=1}^{T}\left( x_i(t) - \hat{x_i} \right)\left( x_j(t) - \hat{x_j} \right)}{\sqrt{\sum_{t=1}^{T}\left( x_i(t) - \hat{x_i} \right)^2\sum_{t=1}^{T}\left( x_j(t) - \hat{x_j} \right)^2}}$
* Eq 2. Should not it be:  $ p_{ij} = \frac{\sum_{k}\left( \rho_{ik} - \hat{\rho_i} \right)\left( \rho_{jk} - \hat{\rho_j} \right)}{\sqrt{\sum_{k=t}\left( \rho_{ik} - \hat{\rho_i} \right)^2\left( \rho_{jk} - \hat{\rho_j} \right)^2}} $
* Table 1. looks very overwhelming, and the colours do not help.
* The whole paper sometimes feels like a sequence of random statements not connected to the rest, especially section 2.7.

---

### Official Review · Reviewer_V5iX · 2024-10-02
**The paper presents a promising approach to classifying schizophrenia and bipolar disorder using dynamic high-order functional connectivity and divisive normalization, which could offer improved diagnostic accuracy. However, the study would benefit from clearer explanations of parameter selection, better visualizations, addressing the imbalance in sensitivity and specificity, and more clarity on the contribution of divisive normalization versus dynamic connectivity in the performance boost.**

**Rating:** 6
**Confidence:** 4

**Review:**

The paper introduces a novel approach to classifying schizophrenia and bipolar disorder using dynamic high-order functional connectivity and divisive normalization. This method offers a promising alternative to traditional low-order connectivity analysis, which has limitations in distinguishing these disorders. By leveraging resting-state fMRI and brain dynamics, the study provides valuable insights and may be among the first to apply high-order functional connectivity maps to this classification problem.

1- In Figure 3, it would be beneficial to include visualizations of sample high-order functional graphs for bipolar patients to help visually identify which functional connections differ between schizophrenia and bipolar disorder subjects.

2- Given the lack of ground truth for functional maps in resting-state brain data, a clearer explanation of how the parameters in Equation 6 were selected would be helpful. The paper mentions specific values but does not discuss how they were optimized.

3- The model reports high specificity, AUC, and F1-scores but low sensitivity, indicating that while it performs well for one group, it struggles to identify the other. This imbalance could be addressed by adjusting the classification threshold, re-examining feature extraction, or applying class weighting to improve sensitivity and balance the metrics more effectively.
It would be valuable if the authors could highlight the most important connections for distinguishing bipolar from schizophrenia subjects, which would help interpret the functional connectivity patterns.

4- It is unclear whether the reported performance improvement is due to the inclusion of the divisive normalization step or the superior performance of dynamic versus static high-order functional connectivity. Clarifying this distinction would provide better insight into the method’s effectiveness.

---

### Official Review · Reviewer_UYpM · 2024-10-07
**Review of "Hybrid Dynamic High-Order Functional Correlations and Divisive Normalization for Improved Classification of Schizophrenia and Bipolar Disorder"**

**Rating:** 7
**Confidence:** 3

**Review:**

The paper makes a valuable contribution to the field of psychiatric disorder classification by introducing hybrid methods that combine dynamic high-order functional correlations (dHOFC) with divisive normalization. The methodological rigor is commendable, and the mathematical formulations provided are generally well-grounded in the literature. The results are promising, as the proposed method shows notable improvements over existing approaches.
* Strengths:
    * The inclusion of both dynamic and high-order functional connectivity is innovative and addresses limitations in previous low-order functional connectivity approaches.
    * Divisive normalization enhances the feature representation and demonstrates superior performance in classification tasks, as shown by a significant increase in AUC (up to 96.61%).
    * The dataset used is well-established (B-SNIP), ensuring the robustness of the findings.
* Weaknesses:
    * The paper could benefit from a more thorough description of the parameter tuning process for divisive normalization and its impact on the final results.
    * Although the results are promising, the comparison with deep learning-based methods could be expanded. It would be beneficial to see more direct comparisons with contemporary neural network approaches in psychiatric disorder classification.
Clarity
The paper is clearly structured, and the flow of the narrative is logical. However, some sections could be explained in more detail to improve accessibility for a broader audience.
* Strengths:
    * The figures, especially those presenting the functional connectivity matrices before and after divisive normalization, are clear and helpful for visualizing the improvements introduced by the method.
    * The mathematical derivations are precise and well-referenced, enhancing the technical robustness of the work.
* Weaknesses:
    * The introduction to divisive normalization and its application in functional connectivity is somewhat brief. A more intuitive explanation for readers unfamiliar with this technique would improve the paper's accessibility.
    * Some notation, particularly in the equations for dynamic high-order functional connectivity (Eq. 5), could be better clarified, as the indices and symbols may confuse non-experts.
Originality
The paper offers a novel combination of dynamic high-order functional connectivity and divisive normalization, which has not been extensively explored in the classification of psychiatric disorders. This approach introduces new perspectives on leveraging complex brain network information for diagnosis.
* Strengths:
    * The fusion of dHOFC and DN is a unique contribution and shows potential for broader applications beyond schizophrenia and bipolar disorder classification.
    * The integration of divisive normalization into fMRI data processing is a fresh application of a technique primarily used in image processing, showcasing the cross-disciplinary potential of the method.
* Weaknesses:
    * While novel, the paper could include more discussions about how this approach might generalize to other psychiatric disorders or neurodegenerative diseases. Furthermore, the theoretical justification for why divisive normalization works well in this context could be expanded.
Significance
The potential impact of this research is substantial. If validated with larger datasets and applied to other disorders, this approach could significantly improve diagnostic accuracy in clinical settings.
* Strengths:
    * The improvements in classification accuracy are substantial, especially with the inclusion of divisive normalization, which boosts the AUC to 96.61%.
    * The methodology has the potential to inform the development of new diagnostic tools that use fMRI data for more precise and personalized psychiatric diagnoses.
* Weaknesses:
    * The practical significance could be enhanced by demonstrating the method’s utility in clinical trials or real-world diagnostic settings. At present, the application remains somewhat theoretical without sufficient validation in applied environments.

Pros
1. Innovative integration of dynamic high-order functional connectivity and divisive normalization.
2. Significant improvements in classification accuracy compared to baseline methods.
3. Well-designed experiments using a robust dataset (B-SNIP).
4. Clear and informative visualizations of functional connectivity matrices.

Cons
1. Insufficient explanation of parameter tuning and its impact on the results.
2. Limited comparison with contemporary deep learning methods for psychiatric disorder classification.
3. Certain notational and technical sections could benefit from more clarity and detail for a broader audience.
4. More extensive real-world validation would strengthen the paper’s impact.

Overall Evaluation
This paper presents a strong contribution to the field of psychiatric disorder classification, especially in the application of fMRI data analysis. The combination of dynamic high-order functional connectivity and divisive normalization is innovative and offers promising results. However, the paper would benefit from additional clarity in the technical explanations and further comparisons with state-of-the-art deep learning models. The real-world significance of the findings could also be more thoroughly explored.

---

### Decision · Program_Chairs · 2024-10-10

**Decision:**

Accept

**Comment:**

In light of the reviewers' feedback and relevancy of the submission, we are pleased to accept this paper for presentation at UniReps 2024. We kindly ask the authors to incorporate the reviewers' suggestions and feedback in the final camera-ready version of the manuscript.